# A Cationic Amphiphilic AIE Polymer for Mitochondrial Targeting and Imaging

**DOI:** 10.3390/pharmaceutics15010103

**Published:** 2022-12-28

**Authors:** Junliang Zhou, Haiyang Wang, Wen Wang, Zhiwei Ma, Zhenguo Chi, Siwei Liu

**Affiliations:** PCFM Lab, GD HPPC Lab, School of Chemistry, Sun Yat-sen University, Guangzhou 510275, China

**Keywords:** mitochondrial targeting, mitochondrial fluorescent probe, cationic amphiphilic polymer

## Abstract

Mitochondria are important organelles that play key roles in generating the energy needed for life and in pathways such as apoptosis. Direct targeting of antitumor drugs, such as doxorubicin (DOX), to mitochondria into cells is an effective approach for cancer therapy and inducing cancer cell death. To achieve targeted and effective delivery of antitumor drugs to tumor cells, to enhance the therapeutic effect, and to reduce the side effects during the treatment, we prepared a cationic amphiphilic polymer with aggregation-induced emission (AIE) characteristic. The polymer could be localized to mitochondria with excellent organelle targeting, and it showed good mitochondrial targeting with low toxicity. The polymer could also self-assemble into doxorubicin-loaded micelles in phosphate buffer, with a particle size of about 4.3 nm, an encapsulation rate of 11.03%, and micelle drug loading that reached 0.49%. The results of in vitro cytotoxicity experiments showed that the optimal dosage was 2.0 μg/mL, which had better inhibitory effect on tumor cells and less biological toxicity on heathy cells. Therefore, the cationic amphiphilic polymer can partially replace expensive commercial mitochondrial targeting reagents, and it can be also used as a drug loading tool to directly target mitochondria in cells for corresponding therapeutic research.

## 1. Introduction

Doxorubicin (DOX) is effective against tumor cells, but the systemic application of this drug usually causes severe side effects in other tissues, such as irreversible cardiotoxicity or nephrotoxicity. Therefore, the clinical application of DOX has been limited [1,2,3]. How to effectively deliver drugs to tumor cells in a targeted manner is one of the key issues in the treatment of cancer.

Mitochondria are important organelles that control multiple signaling pathways critical for cell survival and cell death [4]. In recent years, more and more evidence has shown that mitochondria, as a powerful intracellular power source, plays important roles in cellular metabolism, energy production, apoptosis, and signal transmission, and are related to carcinogenesis [5,6,7]. The malignant transformation of tumor cells is often the result of long-term damage to reactive oxygen species (ROS), a toxic by-product of oxidative phosphorylation, and mitochondria are the main source of ROS [8,9,10]. ROS overproduction promotes mutations in oncogenes. When mutated cells survive this oxidative stress, they may emerge as immortalized cells and initiate tumor progression [10,11]. There is evidence that ROS can mediate the abnormal activity of human telomerase reverse transcriptase (hTERT) in cancer cells. In addition, hTERT can activate telomerase activity, maintain telomere length, prevent telomere degradation, and induce cell immortalization, which are the main hallmarks of cancer [12,13,14]. Therefore, mitochondria are reasonable targets for anticancer therapy. A drug such as doxorubicin is a broad-spectrum antineoplastic agent, known for its accumulation in the nuclear compartment and inhibition of topoisomerase II upon intercalation into DNA [15]. Using mitochondria as a surrogate target for DOX can extend the therapeutic curve and overcome the effects of multidrug resistance (MDR). In the meanwhile, its mitochondria-specific delivery has been shown to exert cytotoxic effects through insertion into mitochondrial DNA or through oxidative damage to DNA, membrane-bound proteins, and enzymes. Inhibition of P-gp significantly reduced drug efflux and mitochondrial damage led to activation of mitochondria-specific apoptotic pathways in tumor cells [16,17,18,19,20].

So far, numerous research attempts have been made on mitochondria-targeting drug carriers. Because the mitochondria of cancer cells are markedly different from those of normal cells, there are a variety of potential mitochondrial targets [21], for example, mitochondrial membrane potential (ΔΨm), mitochondrial protein import machinery, hexokinase 2 (HK2), ion channels, ROS inducer, mitochondrial autophagy inducer, mtDNA, electron transport chain, etc. Glycyrrhetinic acid can interact with mitochondrial ETC, resulting in decreased mitochondrial membrane potential. Glycyrrhetinic acid (GA)-functionalized graphene oxide (GO) can deliver doxorubicin (Dox) to the mitochondria of cancer cells, improve mitochondrial permeability, and enhance mitochondrial drug absorption [22]. However, the cytoplasm is highly viscous, which hinders the diffusion of targeted drug molecules within the cell. However, the method based on nanotechnology can ensure the integrity of the original form of the drug, which has made remarkable progress in mitochondrial drug transport. For example, the distinguishing feature of lipophilic cations is that their positive charge is delocalized over a large and hydrophobic surface area that can easily pass through the phospholipid bilayer [23], enabling them to accumulate into the mitochondrial matrix in response to membrane potential [24,25,26]. However, the inherent toxicity associated with lipophilic cations has hindered their clinical development, and thus, has limited applications [27,28]. Synthetic peptides by altering lipophilicity and charge composition can also act as mitochondria-targeting carriers, which exhibit strong affinity for mitochondria. However, the main problems of these peptides are their large molecular weight and poor water solubility [29,30].

Usually, drug-loaded micelles formed from amphiphilic copolymers are highly impermeable and can maintain long-term stability in blood circulation, thereby, reducing the side effects of drugs on healthy tissues [31,32]. Based on the above considerations, we designed a fluorescent amphiphilic polymer micelle (PTD) as a drug carrier to load doxorubicin by electrostatic interaction. Its hydrophobic shell can dissolve hydrophobic drugs, while the hydrophilic shell can make the entire micellar assembly readily soluble in aqueous environments. We characterized it by UV-visible and fluorescence spectra.The drug loading, formation mechanism, organelle targeting and cytotoxicity of DOX-loaded PTD were studied. The polymeric micelles exhibited excellent targeting selectivity to mitochondrial organelles, even better than the commercial mitochondrial fluorescent probe, MitoRed, in some aspects. The DOX-loaded micellar system designed in this paper is expected to realize the function of combining diagnostic imaging and therapeutic capabilities.

## 2. Materials and Methods

### 2.1. Materials

MitoRed (Ex/Em = 579 nm/600 nm) and LysoGreen (Ex/Em = 443 nm/505 nm) were purchased from KeyGEN BioTECH (Jiangsu, China); 1X phosphate-buffered saline was purchased from Biosharp (Beijing, China); RPMI Medium 1640 basic, fetal bovine serum (FBS), Trypsin-EDTA (0.25%), and penicillin streptomycin were purchased from Gibco (Beijing, China); doxorubicin hydrochloride (98%) was purchased from Aladdin (Shanghai, China); dimethyl sulfoxide was purchased from Guangzhou Chemical Reagent Factory; and the Cell Counting Kit-8 was purchased from Dojindo (Shanghai, China). All other chemicals were purchased from Guangzhou Chemical Reagent Factory.

### 2.2. Synthesis of the Cationic Amphiphilic Aggregation-Induced Luminescent Polymer, PTD

The cationic polymer, PTD (Figure 1), was synthesized according to the method described in our previous work [33]. In this work, the content of segment dimethyl diallyl ammonium chloride (the cationic segment) was fixed at 0.1 mol% (*m*:*n* = 99.9:0.1).

### 2.3. Synthesis of DOX-Loaded PTD

To load DOX in polymer PTD, first, we desalted the commercially available doxorubicin hydrochloride. A certain amount of doxorubicin hydrochloric acid was dissolved in deionized water, 1–2 drops of dilute ammonia (5%) were added dropwise, and the mixture was repeatedly extracted with dichloromethane. Then, the dichloromethane was removed. The obtained doxorubicin was dissolved in DMSO (10 mL), mixed with 10 mL of PTD phosphate-buffered solution (100 mg PTD), and the mixture was stirred at room temperature for 24 h. The mixture was dialyzed against DMSO for 24 h, and then dialyzed against deionized water for another 48 h (the molecular weight cut-off of the dialysis bag is 8000). After lyophilization, the DOX-loaded PTD, a dark purple product (90 mg), was obtained.

The molecular structure of PTD contains both hydrophilic quaternary ammonium segment and hydrophobic tetraphenyl ethylene group, a typical characteristic group with aggregation-induced emission (AIE) properties (Figure 2). Thus, when placed in an aqueous solution, the hydrophobic segments coalesce to form nucleation, while the hydrophilic segments attach to the surface of the nucleus to form corresponding aggregates. As shown in Figure 1, the purple line represents the tetraphenyl group, which is a hydrophobic chain segment. The black line is the quaternary ammonium group and a hydrophilic chain segment. The red dot is DOX. After dispersing PTD and DOX in an aqueous solution, PTD is positively charged and can bind to negatively charged DOX through hydrogen bonding and electrostatic interaction, and then can be loaded to the hydrophobic tetraphene inner surface to form drug-carrying micelles and protect them from degradation.

### 2.4. In Vitro Cytotoxicity Assay

For the cytotoxicity to 4T1 mouse breast cancer cells, the polymer and DOX-loaded polymer were both prepared in 5 mg/mL stock solutions with phosphate buffer solution. The culture solution (RPMI 1640 with 10% fetal bovine serum and 1% penicillin streptomycin) was prepared into a dispersion liquid according to a certain concentration gradient (0.125, 0.25, 0.5, 1.0, and 2.0 μg/mL). Breast cancer cells were counted and seeded in a 96-well plate, containing 100.0 μL of medium, at a density of 1.2 × 10^5^ cells per well. The plate was incubated overnight for 15 h to observe whether the cells adhered. Then, 1640 culture medium containing different concentrations of polymer or DOX-loaded polymer was added according to the area, and the incubation was continued for another 24 h. Subsequently, all wells were enriched with 100.0 μL solution which contained 10% CCK-8 reagent and 90% RPMI 1640 medium. After 2 h incubation, the cell viability was measured by using a microplate reader at a wavelength of 450 nm. The cytotoxicity to 3T3 mouse fibroblasts was also investigated by using a similar method.

### 2.5. Organelle Staining Experiments

For lysosome staining, the 4T1 mouse breast cancer cells were seeded in a culture dish, and after the cells had grown to about 80% of the bottom, the culture medium was removed and washed three times with PBS buffer. The pre-prepared PTD solution (5 μg/mL) was added and incubated for 12 h. Then, the lysosomal probe solution (LysoGreen) with a concentration of 1 μM was added in the dark and placed in an incubator for staining for 1 h.

For mitochondrial staining, the mitochondrial staining method was similar to lysosome staining, except that the concentration of mitochondrial probe solution (MitoRed) was 0.25 μM and the staining time was 20 min.

## 3. Results

### 3.1. In Vitro Cytotoxicity Assay to 4T1 Mouse Breast Cancer Cells

Biotoxicity data are very important for PTD polymers used as drug carriers. The polymer system loaded with doxorubicin should have strong biotoxicity to cancer cells, and at the same time have no or low biotoxicity to healthy cells. After co-incubating breast cancer cells with different concentrations of PTD and DOX-loaded PTD for 24 h, we could see that the cell viability of PTD polymer reached more than 80% in the concentration range from 0.125 μg/mL to 2.0 μg/mL. However, the biotoxicity of DOX-loaded PTD became stronger and stronger with an increase in its concentration. At the concentration of 1.0 μg/mL, cell viability began to decrease significantly, and when the concentration increased to 2.0 μg/mL, cell viability was greatly reduced to about 10% (Figure 3). Therefore, in the drug loading system, DOX-loaded PTD has a good inactivation effect on cancer cells when the concentration is greater than 2.0 μg/mL.

### 3.2. In Vitro Cytotoxicity Assay to 3T3 Mouse Fibroblasts Treated

However, it is well known that cationic polymers are biologically toxic at higher concentrations [34]. Therefore, we also investigated the biotoxicity of this polymer system to healthy cells. Here, we chose 3T3 mouse fibroblasts treated as the research object. Although free DOX has a strong inactivation effect on healthy cells, very low toxicity of the DOX-loaded PTD could be found in our system. It can be seen from Figure 4 that the cell viability of PTD and DOX-loaded PTD is above 80% in the concentration range from 0.125 to 2.0 μg/mL. We speculate that the weak acid environment of breast cancer cells may promote the release of loaded DOX, which in turn, leads to increased toxicity of the material system and inactivation of cancer cells. However, for healthy cells, the neutral and/or alkaline environment is not easy for DOX to dissociate, and therefore, the material remains consistently low in biotoxicity even at higher concentrations. This also indicates that drug-loaded micelles formed by PTD are very stable.

Based on the previous experimental data, for DOX-loaded PTD, the optimal concentration is 2.0 μg/mL, which can ensure sufficient inactivation efficiency on cancer cells as well as ensure the biological activity on normal healthy cells.

### 3.3. Organelle Targeting Studies

Relevant studies in the literature have reported that cationic nanocarriers enter tumor cells mainly through energy-dependent endocytosis [35,36] or cell membrane perforation mechanisms [37,38]. Some are transported to lysosomes, and some are transported to cytoplasm or other organelles, avoiding lysosomes. Therefore, in order to study the organelle targeting properties of the PTD system, we selected 4T1 mouse breast cancer cells as the in vitro cell model, and carried out the following research work in combination with commercial lysosome and mitochondrial staining reagents. The blue fluorescence is polymer fluorescence, the green fluorescence is commercial lysosomal fluorescence, and the red fluorescence is commercial mitochondrial fluorescence. The more overlapping areas and the darker the color, the better the overlap effect. As can be seen from Figure 5, the polymer with blue fluorescence has a better color overlap effect than the red commercial mitochondria, indicating that it has better targeting properties to mitochondria than lysosomes. In addition, PTD is structurally formed by connecting a hydrophilic positively charged group with a hydrophobic group through a chemical bond, which can selectively target mitochondria under the drive of mitochondrial membrane potential. This further confirms that PTD can target mitochondria. Mitochondrial organelle staining can be efficiently achieved at a very low polymer concentration. In addition, the polymer is easy to prepare and inexpensive, and can replace more expensive commercial mitochondrial probes on some occasions.

## 4. Discussion

The purpose of this article is to design and synthesize a class of cationic AIE polymers for partial replacement of more expensive mitochondrial fluorescent markers and to design them as an effective drug carrier for direct entry into organelles. The introduction of cations is for this purpose. It can cause electrostatic adsorption between it and the negatively charged DOX, and with the help of the mitochondrial targeting properties of the PTD polymer, the DOX could smoothly enter the organelles inside the cell. In previous research, we mainly used this polymer for the visual detection of organophosphorus, and its microscopic morphology has been studied in detail [33,39,40]. The particle size distribution of PTD and DOX-loaded PTD used in this paper is shown in Appendix A and the distribution is relatively uniform. The smaller size PTD facilitates its engulfment by cells and smooth entry into organelle mitochondria. Cytotoxicity experiments have indeed confirmed our speculation.

From the current research results, we cannot directly prove whether DOX is released inside the mitochondria. However, from the cytotoxicity data of the PTD containing DOX, it can be seen that DOX can be released in the cancer cell environment, inhibiting the growth of cancer cells. From the normal cytotoxicity experiments, it can be seen that its toxicity is small, indicating that the release of free state DOX is less. It can also be seen from the drug release curve (Appendix A) that DOX released by the drug loading system in the acidic environment is higher than that in the neutral environment, and the loading system is more stable in the neutral environment. Therefore, we can infer that the acidic environment of cancer cells is conducive to the upspring of DOX from PTD. Using this feature, we can, then, use PTD as a vehicle for DOX drug delivery.

In addition, the commercial mitochondrial fluorescent probe (MitoRed) that we purchased cost about 2000 rmb/400 μg, which is expensive. Whereas the main raw materials for the synthesis of PTD are not too expensive. For example, bromotriphenylethylene is about 500 rmb/100 g, dimethyl diallyl ammonium chloride (DMDAAC) is about RMB 75/500 mL, and (4-aminophenyl) boronic acid is about RMB 600/25 g. Moreover, the yield of the synthesis process is high, and we also have a related Chinese patent [41] for the synthesis route of intermediates, and the final product PTD price is much lower than that of commercialized mitochondrial fluorescent probe. In addition, we could adjust the fluorescence wavelength emitted by the polymer by introducing different substituents to obtain better results.

## 5. Conclusions

In summary, a major problem of current antitumor drug therapy is its broad-spectrum toxicity, which inactivates many normal cells while inactivating tumor cells. To address this problem, in this paper, a cationic amphiphilic AIE polymer system is investigated, which has excellent targeting properties to mitochondria. Once loaded with DOX, the DOX can take advantage of its mitochondrial targeting properties to enter cells by targeting mitochondria. For 4T1 mouse breast cancer cells, due to their inherent weak acidic environment, the loaded DOX can be freed from the loading system inside the cancer cells, thereby, inactivating the cancer cells. When the concentration of DOX-loaded micelles was 2.0 μg/mL, it had a good inactivation effect on cancer cells, and the survival rate of cancer cells was 10%, while at the same time, it was basically non-toxic to normal functional 3T3 mouse fibroblasts treated. The cationic amphiphilic polymer reported in this paper is expected to replace expensive mitochondrial fluorescent probe in some cases, and it also demonstrates a drug-loading tool for cancer therapy.

## Figures and Tables

**Figure 1 pharmaceutics-15-00103-f001:**
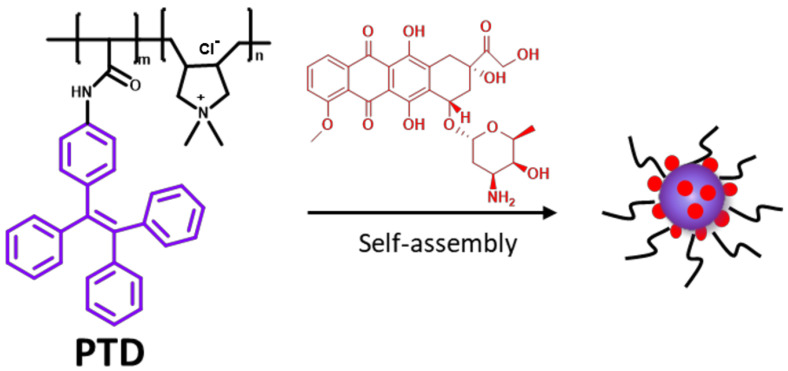
Schematic diagram of PTD assembly and loading DOX in aqueous.

**Figure 2 pharmaceutics-15-00103-f002:**
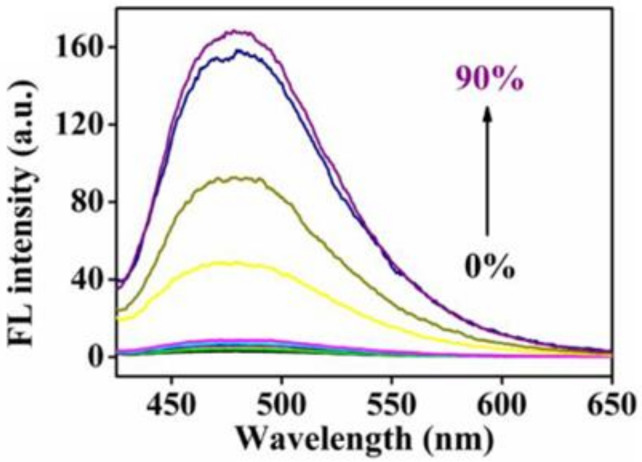
FL spectra of PTD in water-THF mixture with different water fractions, indicating that PTD has typical AIE characteristics.

**Figure 3 pharmaceutics-15-00103-f003:**
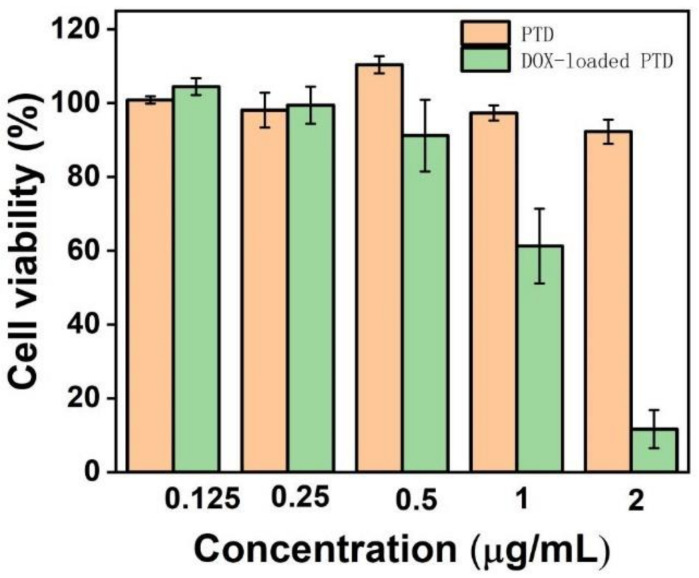
Survival rate of breast cancer cells with different concentrations of blank and DOX-loaded PTD solution within 24 h.

**Figure 4 pharmaceutics-15-00103-f004:**
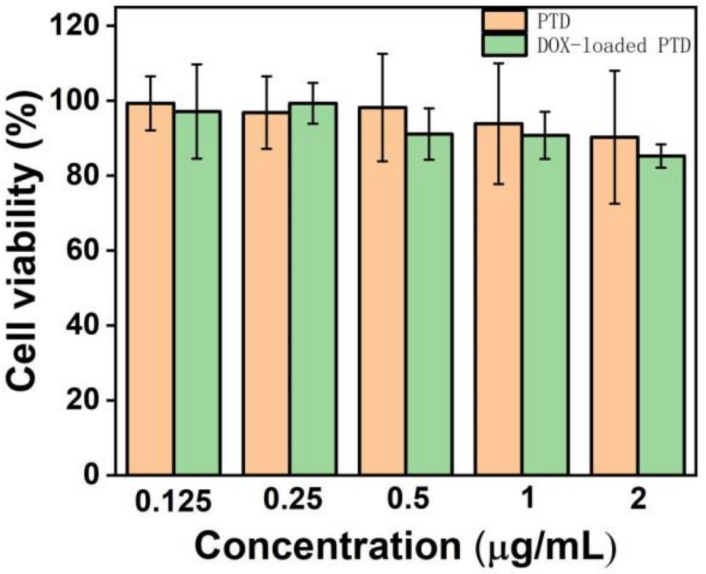
Survival rate of fibroblasts in different concentrations of blank and DOX-loaded PTD solution within 24 h.

**Figure 5 pharmaceutics-15-00103-f005:**
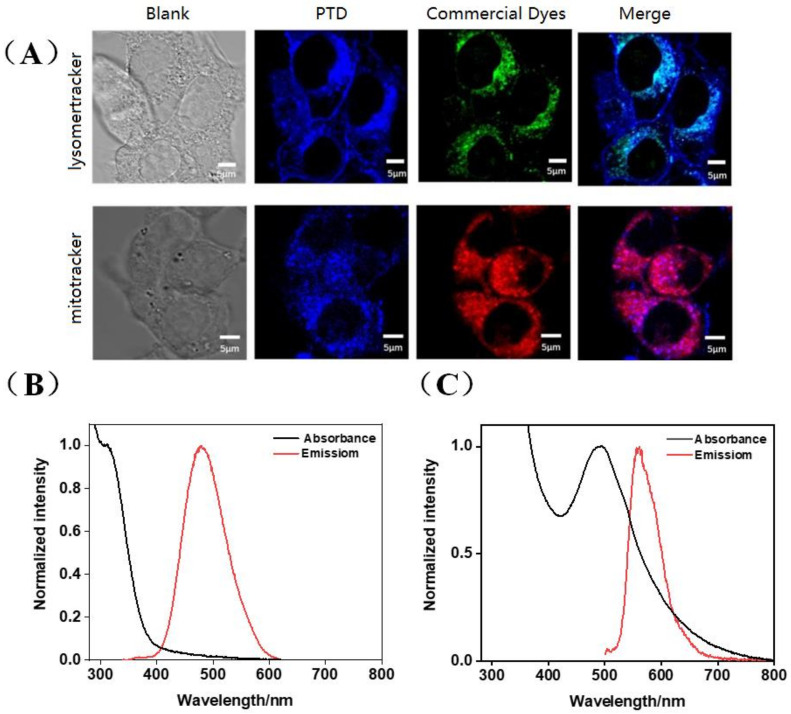
(**A**) The fluorescence image of PTD polymer was observed by confocal laser scanning microscope; (**B**) normalized fluorescence spectrum of PTD; (**C**) normalized fluorescence spectrum of DOX-loaded PTD.

## Data Availability

The data presented in this study are available on request from the corresponding author.

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
