# Peer review of "A Cationic Amphiphilic AIE Polymer for Mitochondrial Targeting and Imaging"

_pharmaceutics, 2022, doi:10.3390/pharmaceutics15010103_

Round 1

Reviewer 1 Report

In this manuscript, the authors described a fluorescent cationic amphiphilic polymer micelle with AIE emission properties as a drug carrier to load doxorubicin. They demonstrated that the micelle can be localized to mitochondria and had a better inhibitory effect on tumor cells than on healthy cells. However, some questions need to be addressed.

1.     The discussion about the AIE emission properties of PTD is not enough. The authors should evaluate the dependence of the fluorescence quantum yield of PTD on the solvent composition of the water/THF mixture. Also, the emission spectra of PTD with different concentrations in pure THF need to be discussed.

2.     Figure 1, the authors claim, “DOX is loaded to the hydrophobic purple interior and surface due to hydrogen bonding and electrostatic interaction, forming drug-loaded micelles.” A more detailed description or experiments should be added to further illustrate the drug-loaded manner.

3.     Figure 2, why does the FL intensity of PTD in the water-THF mixture (90:10, v/v) only show very weak enhancement?

4.     In chapter 3.2, the authors speculate that different pH environments could regulate the release of loaded DOX. In vitro experiments should be added to discuss the stability of DOX-loaded polymer micelle in physiological solutions with different pH.

5.     In organelle targeting studies, overlaid plot profiles of the PTD channel and commercial dyes channel should be added to illustrate the better targeting properties to mitochondria than lysosomes.

6.     Some minor issues: there should be space between the number and the unit; “1.2×105 cells per well” should be replaced by “1.2×105 cells per well”.

Author Response

Thank you for your comments. The relevant replies are attached hereto.

Reviewer 2 Report

I would like to congratulate the authors for putting up this interesting manuscript. being said that I have following suggestions for the authors. 

1. Concept of mitochondrial targeting in cancer therapy have been tried earlier, Authors need to describe more of past studies in the introduction.  

2. In this manuscript authors didn't include any experiments supporting the effect of "PTD and Dox" on mitochondrial health and function. I suggest the authors to include few mitochondrial studies in the manuscript. 

3. In figure 4, Cytotoxicity levels of Dox is almost non-existent, while 2ug almost completely killed the cancer cells (fig 3). Cytotoxicity levels of Dox is known to be very high. Can authors justify their observations?    

4. again, when authors found mitochondrial translocation of polymer, what changes it is bringing the cell physiology, more studies needed to be provided in the manuscript. 

Author Response

(The authors gave the same response as above.)

Reviewer 3 Report

The manuscript proposes a new cationic amphiphilic polymer as a probe or a carrier for mitochondrial targeting. The application is interesting but there are points to be addressed.

Line 61-73 These lines anticipate the obtained results. They do not sound as an introduction.

Line 62-64 It is not clear what the hydrophobic shell and the hydrophilic shell refer to in reference to the architecture of the micelle.

Figure 2 What represents 0% to 90% inside the figure in relation to the presented fluorescence spectra?

The chemical-physical characterization of the doxorubicin loaded micelles is completely missing. What is the size of the loaded micelles? How much is the encapsulation efficiency and doxorubicin drug content?

Line 161-166 Please explain better the reasons about that loaded micelles are more toxic to cancer cells with the respect to the healthy ones. According to my knowledge, release of doxorubicin is favoured at a pH close to 7 with the respect to a lower pH (e.g. pH 5). Some references can also be provided about the discussion of drug release and cytotoxicity of doxorubicin.

Line 178-183 Please discuss better the results present in Figure 5 to support the more efficient staining for mitochiondria than lysosomes. Apparently, larger differences are not evident.

Line 213-214 What is the “luminous color of the polymer”?

The potential use of this cationic amphiphilic polymer as a probe for imaging or as a constituent of micellar carriers for drug mitochondrial targeting should be better defined. I think further experimental studies and evidences are required for both possibile applications.

Author Response

(The authors gave the same response as above.)

Round 2

Reviewer 2 Report

Authors improved the quality of the manuscript to a large extent, however authors didn't include studies showing the mitochondrial health related studies (as suggest in 2nd and 4th comment). For example authors can provide data from mitochondrial membrane permeability studies, seahorse, western for mitochondrial markers, etc. 

Author Response

It has been revised on page 2 of the new version. And we added some data based on comments from other reviewers in supporting information file. Thanks.

Modification: In the meanwhile, their mitochondria-specific delivery has been shown to exert their cytotoxic effects through insertion into mitochondrial DNA or through oxidative damage to DNA, membrane-bound proteins, and enzymes. Inhibition of P-gp significantly reduced drug efflux and mitochondrial damage led to activation of mitochondria-specific apoptotic pathways in tumor cells [16-20].

Reviewer 3 Report

Author answer: It has been revised in the new version. Thanks.

Reviewer: The end of the introduction still sound as a conclusion since it anticipates the obtained results.

Authors answer: The hydrophobic shell is composed of tetraphene groups, while the hydrophilic shell is composed of quaternary ammonium salt groups. We have described it in line 119-121.

Reviewer: Line 124 What is the “purple inner surface”?

Authors answer: The particle size of the drug-loading micelle is ca. 4.3 nm, which has been confirmed in previous studies (Reference 32, 38-39), with the encapsulation rate of 11.03% and drug loading rate of 0.49%. We’ve added them to the abstract of this paper.

Reviewer: I think this information should be also reported in the body of the manuscript, as in the discussion section (Line 216). Anyway, particle size and drug loading should be checked and calculated for any batch of preparation.

Authors answer: DOX was loaded in the hydrophobic core by PTD through hydrogen bonding and electrostatic interaction. The drug-loaded micelles are relatively stable in a neutral environment, making DOX difficult to release. It can also be seen from the drug release curve that DOX is easy to release DOX in an acidic environment. Therefore, the toxicity of DOX-loaded micelles to cancer cells is greater than that of healthy cells due to the acidic cellular environment of cancer cells.

Reviewer: The reason about that loaded micelles are more toxic to cancer cells with the respect to the healthy ones has been explained in the reviewer comments but it is not still clear in the manuscript. Please provide more details in the manuscript or as a supplementary information file.

Author Response

Responses to relevant questions can be found in the attachment. And we've supplemented some data in the supporting information file. Thanks.

Round 3

Reviewer 3 Report

The manuscript is suitable for publication